# Enhanced Photocatalytic Hydrogen Production Activity by Constructing a Robust Organic-Inorganic Hybrid Material Based Fulvalene and TiO_2_

**DOI:** 10.3390/nano12111918

**Published:** 2022-06-03

**Authors:** Mengyuan Wang, Shizhuo Su, Xin Zhong, Derui Kong, Bo Li, Yujie Song, Chunman Jia, Yifan Chen

**Affiliations:** Hainan Provincial Key Laboratory of Fine Chemicals, College of Chemical Engineering and Technology, Hainan University, Haikou 570228, China; wangmengyuanb@163.com (M.W.); 20190411310066@hainanu.edu.cn (S.S.); zhongxin20210831@163.com (X.Z.); 20085600210093@hainanu.edu.cn (D.K.); l1178938050@163.com (B.L.); songyujie@hainanu.edu.cn (Y.S.); jiachunman@hainanu.edu.cn (C.J.)

**Keywords:** photocatalysis, titanium dioxide (TiO_2_), tetrathiafulvalene derivatives, organic-inorganic hybrid nanomaterials, sol-gel method

## Abstract

A novel redox-active organic-inorganic hybrid material (denoted as H_4_TTFTB-TiO_2_) based on tetrathiafulvalene derivatives and titanium dioxide with a micro/mesoporous nanomaterial structure has been synthesized via a facile sol-gel method. In this study, tetrathiafulvalene-3,4,5,6-tetrakis(4-benzoic acid) (H_4_TTFTB) is an ideal electron-rich organic material and has been introduced into TiO_2_ for promoting photocatalytic H_2_ production under visible light irradiation. Notably, the optimized composites demonstrate remarkably enhanced photocatalytic H_2_ evolution performance with a maximum H_2_ evolution rate of 1452 μmol g^−1^ h^−1^, which is much higher than the prototypical counterparts, the common dye-sensitized sample (denoted as H_4_TTFTB-5.0/TiO_2_) (390.8 μmol g^−1^ h^−1^) and pure TiO_2_ (18.87 μmol g^−1^ h^−1^). Moreover, the composites perform with excellent stability even after being used for seven time cycles. A series of characterizations of the morphological structure, the photoelectric physics performance and the photocatalytic activity of the hybrid reveal that the donor-acceptor structural H_4_TTFTB and TiO_2_ have been combined robustly by covalent titanium ester during the synthesis process, which improves the stability of the hybrid nanomaterials, extends visible-light adsorption range and stimulates the separation of photogenerated charges. This work provides new insight for regulating precisely the structure of the fulvalene-based composite at the molecule level and enhances our in-depth fundamental understanding of the photocatalytic mechanism.

## 1. Introduction

H_2_, as a high calorific value and zero pollution, has attracted widespread attention [1,2,3]. Photocatalytic water splitting is considered to be one of the significant strategies for converting solar energy into chemical energy without pollution to the environment [4,5,6,7,8,9]. Using semiconductor photocatalysis to split water is an effective clean hydrogen production technology. Among many photocatalytic semiconductor materials, n-type TiO_2_ has become one of the most promising photocatalysts due to its excellent photoelectrochemical performance, low cost, non-toxic and strong redox activity [10,11,12,13]. However, TiO_2_ has a wide band gap and low visible-light absorption ability, and it can only absorb ultraviolet light (about 4% of solar energy) [14]. Simultaneously, photogenerated carriers are easy to recombine, and quantum efficiency is low, which seriously affects the development of TiO_2_ in the field of photocatalytic hydrogen production. Adjusting the morphological structure and the energy band position on TiO_2_, such as doping of metal and non-metal ions [15,16,17], dye sensitization [18,19,20] and semiconductor combination [14,21,22,23], has been carried out to enhance the photocatalytic activity of TiO_2_. Especially, dye-sensitized composites have been applied to utilize visible-light sections for photocatalytic conversion. However, dye-sensitized composites have poor stability and are easy to deactivate due to dyes detaching and degrading from the surface of the semiconductor under light irradiation [24,25]. To remedy the situation, we used the facile sol-gel method to design and synthesize a series of hybrid materials based on functional organic molecules (calixarene or pillararene derivative) and TiO_2_ for photocatalytic H_2_ production from splitting water under visible light irradiation [20,26,27]. Compared with the dye sensitization systems, these hybrid photocatalysts show outstanding photocatalytic stability due to high robust coordination linkage and efficient photogenerated charges separation and transfer from organic dyes to TiO_2_. However, these calixarene or pillararene derivatives still have some disadvantages in the hybrid system, such as weak optical responsiveness ability (an absorption band edge <500 nm), which hinder its development in photocatalysis. Thus, it is essential to exploit functional organic molecules with their extended visible-light absorption wavelength range to achieve efficient solar-energy conversion.

Tetrathiafulvalene (TTF) and its derivatives own unique electron-rich and redox-active properties [28], which commonly work as functional building blocks to construct two-dimensional (2D) and three-dimensional (3D) framework materials (COF or MOF) and organic-inorganic hybrid materials, and have been intensively studied in the fields of photothermal conversion, photocatalysis [29,30,31,32], superconductors [33], sensors [34,35,36] and solar cell [37,38,39]. For example, Zhou et al. reported a composite (Ag@Dy-m-TTFTB) containing Ag nanoparticles and tetrathiafulvalene-based MOFs materials with a wide absorption wavelengths range (200–1000 nm) and efficient NIR photothermal conversion owing to the doping of the partially oxidized TTF ligands (TTF^•+^) and the plasmonic Ag nanoparticles [40]. Lan et al. designed and synthesized 2-dimensional COFs (TTCOF-Zn) based on porphyrin and tetrathiafulvalene, which worked as catalysts for the photocatalytic coupling reaction of CO_2_ and H_2_O, achieving high photocatalytic CO production with nearly 100% selectivity and accompanying with H_2_O photooxidation to O_2_ [41]. Dai et al. synthetized a self-condenses black organic-inorganic polymeric nanomaterials (B-1) assembled by a cyclic titanium-oxo cluster and organic TTF units, which owned multiporous nanostructure and generous electron density on TTF moiety for outstanding dye selection adsorption, and would even work as photocurrent-responsive electrode materials [42]. The studies above indicate that TTF derivatives have a huge potential application in the field of artificial photocatalysis and could greatly embellish the photophysics property of TiO_2_ nanomaterials. However, the synthesis methods for TTF-based composites are still hard and complicated. Therefore, it is absolutely necessary to exploit TTF-based composites with a flexible, simple preparation processed to endow its wide optical spectrum responsiveness range to fully use solar energy and high optical quantum conversion efficiency for photocatalytic reaction.

Herein, we designed and synthesized binary organic-inorganic composite nanomaterials, which combine TTF derivatives with TiO_2_ nanoparticles by a simple sol-gel method. Among them, TTF derivatives are tetrathiafulvalene-3,4,5,6-tetrakis(4-benzoic acid) (H_4_TTFTB) structure, as superior electron donors/acceptors, enabling the formation of intermolecular π-π stacking with relatively short S-S interactions, which could efficiently facilitate charges separation and transfer under illumination [40]. The organic-inorganic composite nanomaterials loaded with a co-catalyst are expected as an excellent photocatalyst owing to some advantages as below: (1) H_4_TTFTB possesses donors-conjugation-acceptors (D-π-A) electric structure, achieving excellent photoresponse ability and promoting effective photogenerated charge separation from excited H_4_TTFTB to TiO_2_ nanoparticles; (2) high photostable and thermostable H_4_TTFTB molecule has multiple -COOH functional groups that can integrate with Ti(OC_4_H_9_)_4_ to come into being steady TiO_2_-O-TTFTB bonds, improving the stability of the hybrid; (3) the composite nanomaterials have a large surface area due to a multiporous nanostructure, which endues the photocatalysts with ample light-response active sites, leading to effective light harvesting and utilizing in the pores and channels. In this work, we confirm the advantage of the composite based on H_4_TTFTB and TiO_2_ with Pt nanoparticles (Pt NPs) as a co-catalyst. We have optimized the activity of photocatalytic H_2_ evolution from water for the hybrid materials with a mass percentage difference of H_4_TTFTB and TiO_2_. Remarkably, the hybrid materials with a 5.0% mass percentage of H_4_TTFTB, denoted as H_4_TTFTB-TiO_2_-5.0, exhibit the best photocatalytic H_2_ evolution activity with a rate of 1452 μmol g^−1^ h^−1^ (TON = 1101). Moreover, the photocatalytic mechanisms of the H_4_TTFTB-TiO_2_ composite have been explored. It testifies that the outstanding photocatalytic performance of H_4_TTFTB-TiO_2_ composite is attributed to a large surface area, suitable energy band position and multifunctional synergistic catalysis between H_4_TTFTB and TiO_2_. 

## 2. Materials and Methods

### 2.1. Materials

The chemical reagents used in this work were of analytic grade and were used without further purification. Tetrabutyl titanate (Ti(OBu)_4_, ≥99.0%), acetic acid (CH_3_COOH, ≥99.5%), N,N-dimethylformamide (DMF, 99.9%), tetrahydrofuran (THF, 99.9%) and triethanolamine (TEOA, >99.0%) were purchased from Innochem company (Beijing, China). Tetrathiafulvalene-3,4,5,6-tetrakis(4-benzoic acid) (H_4_TTFTB, 98%) was purchased from Chemextension company (Jilin, China).

### 2.2. Synthesis of Photocatalysts

#### 2.2.1. Synthesis of H_4_TTFTB-TiO_2_ Materials

The structure of H_4_TTFTB and H_4_TTFTB-TiO_2_ hybrid materials are shown in Appendix A, respectively. To prepare H_4_TTFTB-TiO_2_-5.0, THF (2.0 mL), acetic acid (336 μL), distilled water (212 μL) and Tetrabutyl titanate (2 mL) were subsequently added to a 20 mL glass sample bottle, and then H_4_TTFTB (24.0 mg) in DMF (2 mL) was added. The color of the mixture changed from light yellow to dark red and, when stirred, became transparent gelatum. The gelatum was heated at 75 °C overnight to obtain a red solid. The residue was soxhleted with distilled water at 100 °C for 48 h and evaporated to dryness. The hybrid nanomaterials were calcined in a Muffle furnace at 250 °C for 2 h to become yellow-brown powder. Similarly, the hybrid material was handled as the above process with different calcination temperatures (150 °C, 250 °C and 350 °C) and then denoted as H_4_TTFTB-TiO_2_ (150 °C), H_4_TTFTB-TiO_2_ (250 °C) and H_4_TTFTB-TiO_2_ (350 °C). In order to elaborate the formation process of the hybrid materials H_4_TTFTB-TiO_2_, the intermediate products at various reaction times were collected to characterize their crystalline states by checking their XRD patterns (Appendix A). 

H_4_TTFTB-TiO_2_ materials with different H_4_TTFTB mass contents (0, 1.7%, 3.4% and 6.5%) were prepared with the above method and denoted as *a*-TiO_2_, H_4_TTFTB-TiO_2_-1.7, H_4_TTFTB-TiO_2_-3.4 and H_4_TTFTB-TiO_2_-6.5, respectively.

#### 2.2.2. Synthesis of Dye-Sensitized Composites

The *a*-TiO_2_ powder was dispersed in 0.7 mM DMF solution of H_4_TTFTB at room temperature in the dark for 6 h and then centrifuged and washed with DMF and methanol until no color was observed from the eluent. The solid was dried under a vacuum oven at room temperature, denoted as H_4_TTFTB-5.0/TiO_2_.

#### 2.2.3. Synthesis of Composites Materials Loaded with Pt NPs 

The hybrid material above (20 mg) was dispersed in a photocatalytic reaction cell containing 4 mL H_2_O and 10 mL methanol. K_2_PtCl_4_ aqueous solution (0.01 M) was added under stirring, and then the resulting mixture was stirred another 1 h under full-light irradiation by a 300 W xenon lamp. After illumination, the material was centrifugally washed with water and methanol and then dried in a vacuum. Pt NPs’ actual loading amounts tested by inductively coupled plasma atomic emission spectrometry (ICP-AES) were 0.39 wt%, 0.88 wt% and 1.0 wt% (Appendix A).

### 2.3. Photocatalytic Hydrogen Production

The photocatalytic hydrogen evolution by water splitting was carried out in a glass reaction cell with a quartz cover connected to a closed gas circulation, which was swept by high-purity Ar before illumination. A total of 20 mg photocatalysts were dispersed in 20 mL of 10 vol% TEOA aqueous solution. Then the suspension was exposed to a 300 W Xe lamp equipped with an optical filter (λ > 400 nm) to cut off the light in the ultraviolet region. The reaction solution was stirred continuously and cooled to maintain a suitable temperature by the circulation of cooling water. The amount of hydrogen evolved was determined at an interval of 1 h with online gas chromatography. 

### 2.4. Materials Characterization 

The solid ultraviolet diffraction spectrum was measured using a Shimadzu UV-2600 spectrometer (Shimadzu, Kyoto, Japan), and fluorescence spectra were measured by a Fluorolog-3 spectrometer (HORIBA, Shanghai, China). The FT-IR spectra were monitored by a Nicolet iS50 spectrometer (Thermo Fisher Scientific, Waltham, MA, USA). Powder X-ray diffraction (PXRD) was recorded on a Rigaku Smart Lab (Rigaku, Beijing, China) diffractometer (Bragg-Brentano geometry, Cu-Kɑ1 radiation, λ = 1.54056 Å). Transmission electron microscopy (TEM) images, high-resolution transmission electron microscopy (HRTEM), high-angle annular dark-field scanning transmission electron microscopy (HAADF-STEM) micrographs, selected area electron diffraction (SAED), energy-dispersive X-ray spectroscopy (EDX) and elemental mapping were obtained by an FEI Talos F200X (FEI, Hillsboro, OR, USA) transmission electron microscope at 200 kV. Scanning electron microscope (SEM) micrographs were recorded on a SU8020 (HITACHI, Tokyo, Japan)ultrahigh resolution field emission scanning electron microscope. Gas adsorption measurements were performed using ultra-high purity N_2_ on the Quantachrome Auto-sorb-iQ2-MP analyzer (Quantachrome, Beijing, China). X-ray photoelectron spectroscopy (XPS) (Thermo escalab 250Xi, Thermo Fisher Scientific, Waltham, MA, USA) measurements were carried out using a monochromated Al Kα X-ray source (hν = 1486.6 eV) with a 400 μm spot size in an ultrahigh vacuum chamber. The pass energy was 30 eV. Thermogravimetric (TG) analysis was performed at a constant heating rate of 10 °C min^−1^ from 30 to 600 °C in air, on a NETZSCH STA 449 F5/F3 Jupiter thermal analyzer (NETZSCH, Free State of Bavaria, Germany). The collection system of photocatalytic hydrogen production was Perfect Light Lab Solar-6A (PerfectLight, Beijing, China), and H_2_ measurement was performed on FuLi GC9790Ⅱ (FuLi, Taizhou, China). The amounts of Pt nanoparticles loaded on samples were detected by Agilent ICP-OES730 (Agilent, Palo Alto, CA, USA).

### 2.5. Photolectrochemical Methods

The Mott–Schottky curves were conducted on an Ivium-n-Stat electrochemical analyzer (Ivium Technologies B.V., Eindhoven, Netherlands) in a three-electrode cell. A platinum plate was used as a counter electrode, and an Ag/AgCl electrode (3 M KCl) was used as a reference electrode. The electrolyte was a 0.5 M Na_2_SO_4_ solution. In order to prepare the working electrolyte, 5 mg samples were added to a mixed solution of ethanol and Nafion. The photocurrent response and electrochemical impedance measurements were performed using a 0.5 M Na_2_SO_4_ solution electrolyte, using a 300 W xenon lamp with a 400 nm cut-off filter as the light source. Cyclic voltammograms (CVs) were recorded using an Ivium-n-Stat electrochemical analyzer with the platinum plate as the working electrode, Ag/AgCl as the reference electrode and Pt wire as the counter electrode. CVs were measured using 0.1 M tetrabutylammonium hexafluorophosphate (TBAPF_6_) as a supporting electrolyte in DMF with a scan rate of 50 mV s^−1^.

## 3. Results and Discussions

Figure 1a,b display the XRD patterns of *a*-TiO_2_ and H_4_TTFTB-TiO_2_(1.7/3.4/5.0/6.5) hybrid materials. The obtained diffraction peaks at 25.54°, 37.85°, 47.90°, 54.22° and 62.70° were consistent with the standard card of anatase TiO_2_ (JCPDS 21–1272), which demonstrated the formation of anatase TiO_2_ (Figure 1a). Calcination at 250 °C can remove impurities and facilitate the separation and transfer of photogenerated carriers. Simultaneously, the XRD analysis of H_4_TTFTB-TiO_2_ hybrid materials calcined at 250 °C is shown in Figure 1b. As can be seen (Appendix A), no obvious diffraction peaks of H_4_TTFTB can be found, and it shows little impact on the crystallinity due to low contents and high dispersion of H_4_TTFTB in H_4_TTFTB-TiO_2_ hybrid nanomaterials.

Figure 2 shows the N_2_ adsorption-desorption isotherms at 77K of *a*-TiO_2_ and H_4_TTFTB-TiO_2_-5.0 materials. The isotherms plots have a typical type IV pattern, which is evidently characteristic of mesoporous materials. The prepared *a*-TiO_2_ and H_4_TTFTB-TiO_2_-5.0 possess high BET specific surface area, 184.1 m^2^/g and 212.7 m^2^/g, respectively. These results show that the introduction of H_4_TTFTB could appropriately increase the specific surface area of the hybrid nanomaterials. Moreover, the pore size was simulated and calculated by the Barrett-Joyner-Halenda (BJH) method. From the pore size distribution diagram (Appendix A), it could be observed that the pore size distribution (3.630 nm) of H_4_TTFTB-TiO_2_-5.0 nanomaterial is smaller than that (4.411 nm) of *a*-TiO_2_. The detailed data of the surface areas and the pore sizes are summarized in Appendix A. In general, H_2_ production involves surface adsorption and interface reaction of reaction substrates in the heterogeneous photocatalyst system, which could be significantly affected by the morphology and structure of the photocatalyst. Thus, H_4_TTFTB-TiO_2_ composite nanomaterials prepared through a sol-gel process endow mesoporous hierarchical structure and multifunctional synergistic effect, which results in broad visible-light response and more efficient photocatalytic performance.

The morphology and structure of the hybrid were characterized by SEM, TEM, HRTEM and SAED, as well as the element mapping. The SEM image of H_4_TTFTB-TiO_2_-5.0 material is recorded in Appendix A, showing the size of ellipse nanoparticles as about 10–15 nm. To enhance the activity of photocatalytic H_2_ production, H_4_TTFTB-TiO_2_-5.0 was loaded with Pt NPs by the photodeposition method. TEM images of Pt@H_4_TTFTB-TiO_2_-5.0 show that platinum particles are uniformly distributed, and the size is about 2–5 nm (Figure 3a). The HRTEM image indicates that the distance of two adjacent lattice planes of Pt NPs is about 0.226 nm, consistent with the spacing of Pt NPs (111) planes (Figure 3b). The SAED result proves the crystallinity of H_4_TTFTB-TiO_2_-5.0 nanomaterials (Figure 3c). The HAADF-STEM mapping and EDX of Pt@H_4_TTFTB-TiO_2_-5.0 certify the existence of H_4_TTFTB, TiO_2_ and Pt NPs in the composite and C, O, Ti, S and Pt are uniformity dispersed throughout all of the composites (Appendix A and Figure 3d–i, Appendix A). In addition, XPS measurements for *a*-TiO_2_, H_4_TTFTB-TiO_2_-5.0 and Pt@H_4_TTFTB-TiO_2–_5.0 were carried out (Appendix A). Generally, pollution carbon C1s (284.8 eV) is used for charge calibration for all samples. In Appendix A, the survey spectrum displays that C, O, Ti and S exist in the H_4_TTFTB-TiO_2_-5.0. The binding energy of Ti 2p_3/2_ and Ti 2p_1/2_ is at 458.7 and 464.4 eV, respectively, and the binding energy of O 1s is at 529.9 eV, which could be attributed to the Ti-O bond of TiO_2_ for *a*-TiO_2_ and H_4_TTFTB-TiO_2_-5.0 (Appendix A). Moreover, the compositions of Ti and O for both *a*-TiO_2_ and H_4_TTFTB-TiO_2_-5.0 display the expected 1:2 ratio, indicating the existence of TiO_2_ in the hybrid. Compared to *a*-TiO_2_, the binding energy at 164.2 eV and 165.2 eV, 168.6 eV and 169.6 eV belongs to S 2p_3/2_ and S 2p_1/2_, respectively, demonstrating that H_4_TTFTB is successfully doped within TiO_2_. Furthermore, the two different types of peaks represent the existence of the neutral sulfur and oxidized sulfur of H_4_TTFTB in the composite (Appendix A). However, the partially oxidized sulfur of H_4_TTFTB in the composite indicates that the electron-rich tetrathiafulvalene (TTF) could be oxidized easily to its stable radical oxidation states (TTF^•+^) during the synthesis process of the hybrid. Furthermore, when the composite H_4_TTFTB-TiO_2_-5.0 was loaded with the Pt NPs by the photoreduction with the aid of methanol, the proportion of neutral sulfur of H_4_TTFTB in the hybrid would increase owing to the reduction of the oxidized sulfur by methanol. In addition, the banding energy at 71.1 eV and 74.4 eV belongs to Pt 4f_7/2_ and Pt 4f_5/2_, respectively, which could be attributed to Pt (0), also the weak banding energy at 72.5 eV (Pt 4f_7/2_) and 75.6 eV (Pt 4f_5/2_), which could be attributed to a small amount of unreduced Pt^2+^ (Appendix A). 

The FT-IR spectra of *a*-TiO_2_, H_4_TTFTB and H_4_TTFTB-TiO_2_-5.0 hybrid are displayed in Figure 4. Notably, 1621 cm^−1^ and 3417 cm^−1^ of absorption peaks are attributed to the bending and stretching vibrations of -OH groups from *a*-TiO_2_, which is pivotal for the combination with H_4_TTFTB molecules. The stretching vibrations of C=O groups and C-O groups from H_4_TTFTB are at 1696 cm^−1^ and 1103–1378 cm^−1^, respectively. Moreover, the characteristic vibration bands of the benzene skeleton are around 650–900 cm^−1^, which conforms to the structure feature of H_4_TTFTB. Compared with *a*-TiO_2_, the characteristic vibration bands of C=C from H_4_TTFTB and Ti-O from *a*-TiO_2_ appear at 1500 cm^−1^ and 500–700 cm^−1^, respectively, indicating the existence of H_4_TTFTB and TiO_2_ in the hybrid materials. Furthermore, the C=O stretching vibration of H_4_TTFTB at 1696 cm^−1^ disappears, and two new bands appear at 1407 cm^−1^ and 1621 cm^−1^ in H_4_TTFTB-TiO_2_-5.0, which could be attributed to the symmetric and anti-symmetric stretch of -COO-. Thus, H_4_TTFTB should be chemically combined with TiO_2_ through titanium ester bonds (O=C-O-Ti), which could also enhance the separation and transfer of photogenerated charges between H_4_TTFTB and Ti (3d) orbital manifold of *a*-TiO_2_ [43,44]. Further, thermogravimetric analysis (TGA) of H_4_TTFTB and of H_4_TTFTB-TiO_2_-5.0 were carried out to evaluate their thermostability, revealing that the organic molecular H_4_TTFTB and the hybrid materials could be stable up to 300 °C (Appendix A).

The UV-vis diffraction spectra of H_4_TTFTB, H_4_TTFTB-TiO_2_-5.0 and *a*-TiO_2_ are displayed in Figure 5a. As we can see, H_4_TTFTB shows a strong absorption range of about 300–650 nm, ascribing to n-π* or π-π* transition of the phenyl-TTF moiety. The small shoulder at around 650–800 nm could be attributed to the partially oxidized TTF ligands (TTF^•+^) [40]. The UV-vis absorption spectrum and the emission spectra of H_4_TTFTB are exhibited in Appendix A. The maximum emission wavelength of H_4_TTFTB is located at 650 nm; thus, its transition energies (E_0–0_) are calculated to be 2.34 eV. However, *a*-TiO_2_ only has a strong UV absorption peak with a 400 nm absorption edge, which matches the verified band gap (about 3.2 eV) (Figure 5b) [12]. H_4_TTFTB is reacted with tetrabutyl titanate in situ by sol-gel method to obtain composite H_4_TTFTB-TiO_2_. The composite has a broad absorption range at about 300–600 nm, along with a smaller shoulder adsorption peak at 650–800 nm, owing to the low contents of H_4_TTFTB doped. Compared with pure TiO_2_, the H_4_TTFTB-TiO_2_-5.0 composite has two band gaps: a wide band gap of 2.8 eV and a narrow band gap of 2.0 eV (Figure 5b), attributed to *a*-TiO_2_ and H_4_TTFTB, respectively. The band gap of H_4_TTFTB in the composite is lower than the E_0–0_ value of 2.34 eV in pure H_4_TTFTB, indicating that a robust combination exists between H_4_TTFTB and TiO_2_ in the hybrid. 

In order to elucidate the photocatalytic mechanism of H_4_TTFTB-TiO_2_ under visible light irradiation, Mott–Schottky (M-S) measurements of *a*-TiO_2_ were carried out at different frequencies of 500 Hz, 1000 Hz and 1500 Hz (Appendix A). The positive slopes of the M-S plots match with an n-type semiconductor. The flat band positions (V_fb_) of *a*-TiO_2_ are calculated to be −0.7 V (vs. Ag/AgCl). As well known, the conduction band (CB) of n-type semiconductors is approximately the same as the value of V_fb_ [25]. Consequently, the CB of *a*-TiO_2_ is about −0.7 eV (vs. Ag/AgCl), being more negative than the redox potential of H^+^/H_2_, thus, favoring the photoreduction of the proton from water to produce H_2_. The valence band (VB) of *a*-TiO_2_ is counted to be 2.5 eV (vs. Ag/AgCl). Moreover, the first oxidation potential of H_4_TTFTB was detected by CVs characterization to acquire the highest occupied molecular orbital (HOMO) value (Appendix A). The redox potential of excited H_4_TTFTB* is about −1.64 eV (vs. Ag/AgCl); namely, the bottom of the lowest unoccupied molecular orbital (LUMO) is counted from the HOMO and the E_0–0_ value of H_4_TTFTB, which is more negative than the CB of *a*-TiO_2_. Thus, it is thermodynamically favorable for the transfer to be directly from the H_4_TTFTB* to CB of *a*-TiO_2_. 

The photocatalytic activities of H_4_TTFTB-TiO_2_-5.0 catalysts with different amounts of H4TTFTB-doping and with different contents (0.5/1.0/1.5 wt%) of Pt NPs loading have been optimized under visible light irradiation. A series of experiment data are presented in Figure 6a. The results demonstrate that H_4_TTFTB-TiO_2_-5.0 with 1.0 wt% Pt NPs has the optimized photocatalytic activity (Appendix A). Moreover, the photoactivity increases with the addition of H_4_TTFTB-doped amount (1.7 wt%, 3.4 wt% and 5.0 wt%), reaching a maximum for H_4_TTFTB-TiO_2_-5.0. However, further increasing the percentage of H_4_TTFTB to 6.5 wt% with a decrease in photoactivity indicates that the photoactivity of H_4_TTFTB-TiO_2_ nanomaterials could not correlate only to H_4_TTFTB amounts. Reasonable speculation is that excessive H_4_TTFTB in the composite could block porous channels and impede electron migrating and transporting. The photocatalytic activity of the composites nanomaterials is influenced by many factors, such as temperature, pH, solvents, additives, etc. Among them, the calcination temperature is pivotal in building up the structural morphology of composites and enhancing the photocatalytic activity. Thus, the calcination temperature of H_4_TTFTB-TiO_2_-5.0 was optimized. In addition, H_4_TTFTB-TiO_2_-5.0 with 250 °C calcination exhibits the best photocatalytic rates of H_2_ production, which is greatly superior to higher or lower calcination temperature (150 °C or 350 °C) (Appendix A). It is believed that appropriate calcination temperature not only could eliminate the impurity but also improve the crystallinity of the hybrid materials. However, a lower temperature (150 °C) could not meet the requirements for the synthesis of the hybrid, and a higher calcination temperature (350 °C) could destroy the organic component H_4_TTFTB.

In contrast, the rate of photocatalytic H_2_ production of *a*-TiO_2_ and sensitization H_4_TTFTB-5.0/TiO_2_ with only 1.0 wt% Pt (0) loading are 18.87 and 390.8 μmol g^−1^ h^−1^ which are both less than the rates of H_4_TTFTB-TiO_2_-5.0. It demonstrates that the composites have excellent photocatalytic activity than the dye sensitization TiO_2_ system owing to the more efficient migration of photogenerated electrons from H_4_TTFTB* to TiO_2_ through the robust linkages between H_4_TTFTB and TiO_2_. Moreover, to assess the stability of H_4_TTFTB-TiO_2_ nanomaterial, we conducted photocatalytic H_2_ evolution experiments at intervals of 5 h. The H_4_TTFTB-TiO_2_ composite was reused by centrifugation, washed with water and methanol and then vacuum dried. As shown in Figure 6b, the photocatalytic activity of H_4_TTFTB-TiO_2_ shows no significant decrease, revealing high photocatalytic stability in photocatalytic reducing H_2_ with seven recycles. The TON_H2_ is calculated as the mole numbers of H_2_ over the mole numbers of Pt NPs, giving significant results of 1101 after 35 h of photoreactions. In general, compared with other reported photocatalysts (Appendix A), the photocatalytic activity of the H_4_TTFTB-TiO_2_-5.0 system is moderate. However, the simple and flexible synthesis approach has been proposed in this study to obtain the robust fulvalene-based composite, which is significant to shed light on the rational design of photocatalysts and understand the photocatalytic inherent nature. Thus, our next work will focus on regulating precisely the structure of the fulvalene-based composite at a molecule level, which would enhance our in-depth fundamental understanding of the photocatalytic mechanism.

The photocurrent-time (i-t) curves and the electrochemical impedance spectroscopy (EIS) of *a*-TiO_2_, H_4_TTFTB-5.0/TiO_2_ and H_4_TTFTB-TiO_2_-5.0 have been monitored under visible light illumination to further verify the influence of enhanced photoresponse on the photocatalytic process (Figure 6c,d). It is clear that the photocurrent of H_4_TTFTB-TiO_2_-5.0 is higher than H_4_TTFTB-5.0/TiO_2_, indicating that effective interfacial electrons migration and transfer between H_4_TTFTB and TiO_2_ in H_4_TTFTB-TiO_2_-5.0 (Figure 6c). Figure 6d displays the EIS Nyquist plots for samples. Compared with *a*-TiO_2_ and H_4_TTFTB-5.0/TiO_2_, H_4_TTFTB-TiO_2_-5.0 exhibits a smaller semicircular diameter in Nyquist plots, suggesting that the smaller resistance of photogenerated charge migration and transfers on the electrode surface. Thus, the result demonstrates that the hybrid materials possess a smaller interfacial charge-transfer resistance, which could significantly enhance the photoactivity for H_2_ generation, hinting that more effective interfacial photogenerated electrons migrate and transfer between H_4_TTFTB and TiO_2_ in H_4_TTFTB-TiO_2_ composites.

According to the results and discussions above, a possible photocatalytic mechanism for the H_4_TTFTB-TiO_2_ catalytic system is proposed, as shown in Figure 7. H_4_TTFTB owns excellent light absorbance ability to act as a light harvester under visible light irradiation. The excited electrons jump from HOMO into LUMO of H_4_TTFTB and inject into the CB band of TiO_2_. The electrons separated successfully could reduce proton to H_2_ production with the aid of Pt (0) NPs. Herein, Pt (0) NPs are loaded on the surface of H_4_TTFTB-TiO_2_ by the in-situ photodeposition and commonly work as co-catalysts to provide reactive sites, owing to the formation of the Schottky barrier at the interface of Pt NPs and the hybrid, which could be beneficial to promote the separation and transfer of photogenerated charges and enhance the photocatalytic activity. The electrons on the CB of *a*-TiO_2_ could be captured by the co-catalysts and then efficiently photoreduce H^+^ to H_2_. The holes retained in the VB of H_4_TTFTB are neutralized by sacrificial reagent triethanolamine (TEOA). Furthermore, the oxidative H_4_TTFTB^+^ could also be reduced by TEOA to obtain relative H_4_TTFTB.

## 4. Conclusions

In summary, we have in situ synthesized organic-inorganic H_4_TTFTB-TiO_2_ composite containing organic light-harvesting unit H_4_TTFTB and inorganic semiconductor TiO_2_ by the sol-gel method. H_4_TTFTB-TiO_2_-5.0 material exhibits high photocatalytic efficiency under visible light irradiation. A series of characterizations on structural morphology and photoelectric physical and chemical properties suggest that H_4_TTFTB is successfully doped into H_4_TTFTB-TiO_2_, which is attributed to the formation of stable titanium ester bond linkage between H_4_TTFTB and TiO_2_. The absorption ability of H_4_TTFTB extended to visible light range could fully use the solar energy. The multifunctional synergistic effect of organic and inorganic components of H_4_TTFTB-TiO_2_ composite could promote efficient electron migration and transfer in the interface. This work sheds light on the relationship of structure, photocatalytic activity and reaction pathway of TTF-based composite nanomaterials for splitting water to H_2_ production under visible light irradiation.

## Figures and Tables

**Figure 1 nanomaterials-12-01918-f001:**
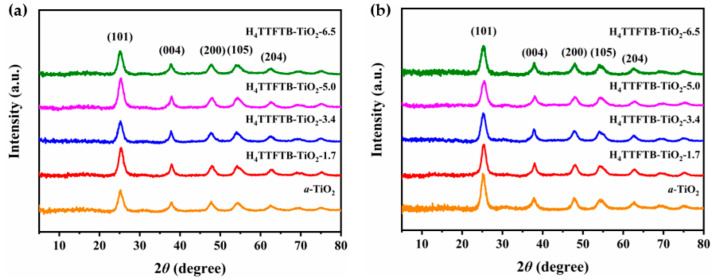
XRD of H_4_TTFTB-TiO_2_ with different H_4_TTFTB amounts were listed above without (**a**) and with (**b**) 250 °C calcination.

**Figure 2 nanomaterials-12-01918-f002:**
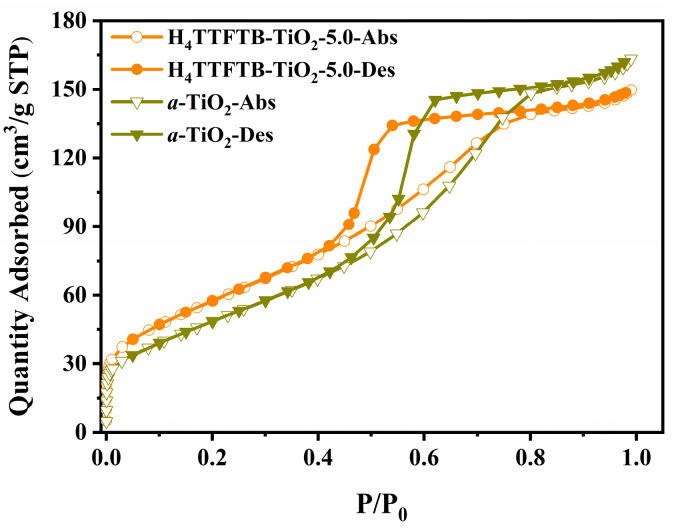
N_2_ adsorption-desorption isotherms of H_4_TTFTB-TiO_2_ system measured under 77 K.

**Figure 3 nanomaterials-12-01918-f003:**
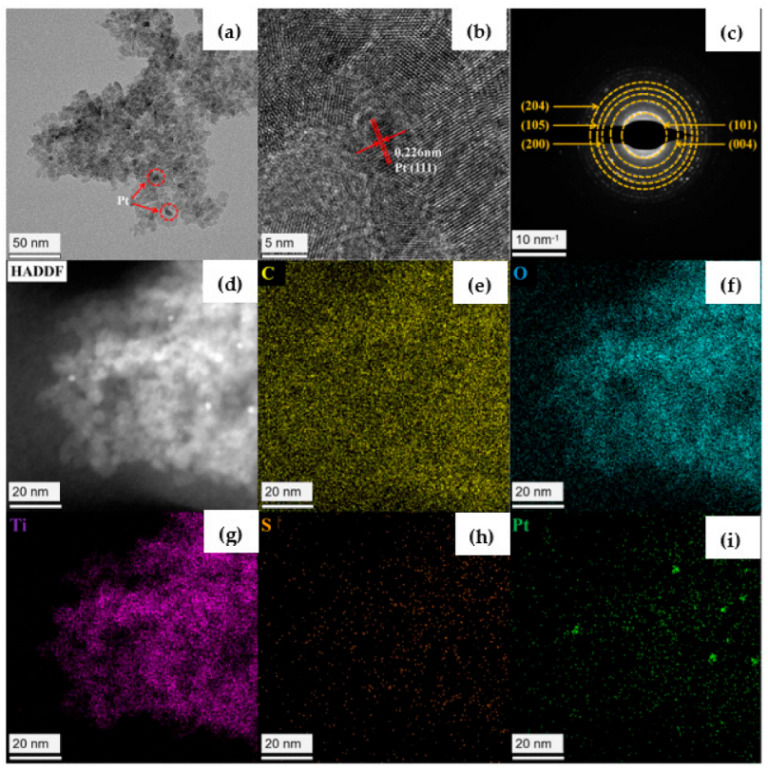
(**a**) TEM, (**b**) HRTEM, (**c**) SAED of H_4_TTFTB-TiO_2_ and (**d**–**i**) the elemental mappings image of Pt@ H_4_TTFTB-TiO_2_-5.0.

**Figure 4 nanomaterials-12-01918-f004:**
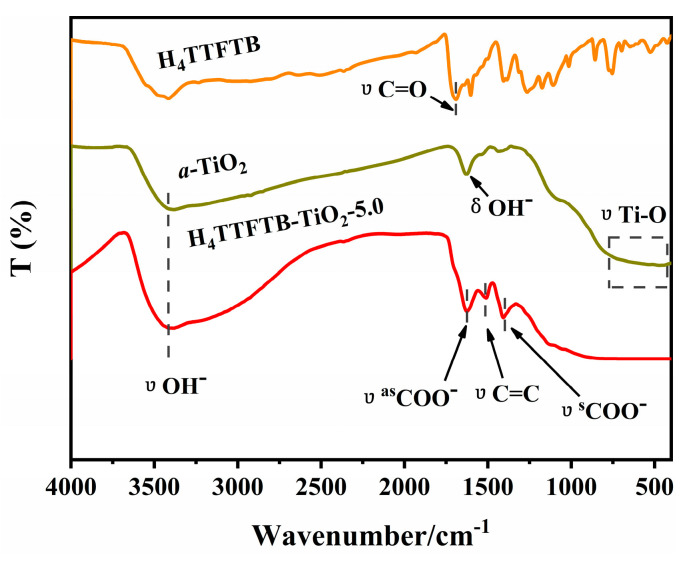
The FT-IR spectra of H_4_TTFTB-TiO_2_-5.0, H_4_TTFTB, *a*-TiO_2_.

**Figure 5 nanomaterials-12-01918-f005:**
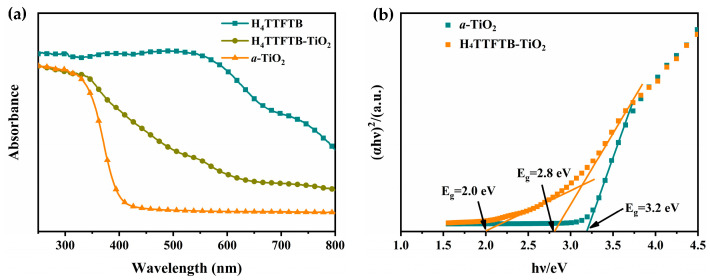
(**a**) UV-vis diffraction spectra of *a*-TiO_2_, H_4_TTFTB and H_4_TTFTB-TiO_2_-5.0, (**b**) the Tauc plot of *a*-TiO_2_ and H_4_TTFTB-TiO_2_-5.0.

**Figure 6 nanomaterials-12-01918-f006:**
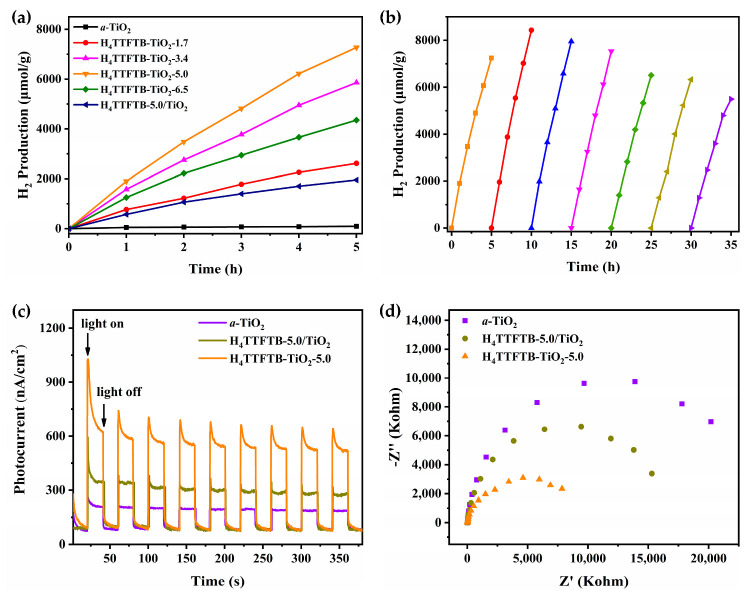
(**a**) Effect of the amount of H_4_TTFTB doped into the TiO_2_ on the performance of photocatalytic hydrogen production, (**b**) Photocatalytic H_2_ production over the recyclability of H_4_TTFTB-TiO_2_-5.0 in 20 mL H_2_O/TEOA (9:1 *v/v*) with 1.0 wt% Pt loading under visible light irradiation (λ > 400 nm), (**c**) Transient photocurrent responses and (**d**) EIS Nyquist plots.

**Figure 7 nanomaterials-12-01918-f007:**
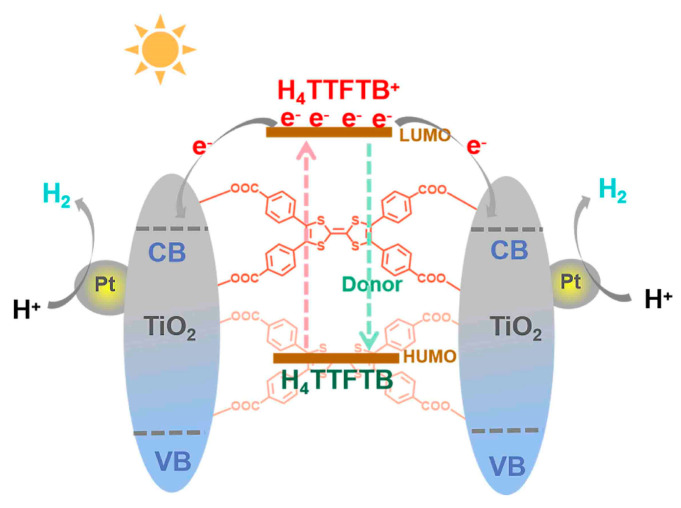
Proposed mechanism of photocatalytic H_2_ production over Pt@H_4_TTFTB-TiO_2_.

## Data Availability

The data presented in this article is available on request from the corresponding author.

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
