# Peer review of "Enhanced Photocatalytic Hydrogen Production Activity by Constructing a Robust Organic-Inorganic Hybrid Material Based Fulvalene and TiO2"

_nanomaterials, 2022, doi:10.3390/nano12111918_

Round 1
Reviewer 2 Report
The manuscript entitled “Enhanced photocatalytic hydrogen production activity by constructing an robust organic-inorganic hybrid material based fuvalene and TiO2”
In title, I believe the authors mean fulvalene, not fuvalene. Also, instead of “an robust”, should be “a robust”
In Figure S1, the structure from H4TTFTB should be included.
Concerning materials characterization, the standard procedures seem to be sufficient to allow clear structural certainty.
It is clear what H4TTFTB-TiO2-5.0 represents, but no so much for H4TTFTB-5.0/TiO2. Please clarify in the text.
Regardless of the results, how does this system compares with literature? There must be nearly hundreds of similar systems appearing every month or week. How does this system compares to that? Against systems that incorporate dyes, even similar to fulvalene derivatives?
This must be cleared in the manuscript.
The manuscript must be carefully checked for inconsistencies in English language. There many problems in fluently reading the manuscript, particularly with gender, tense and number use.
Minor
In Fig S9, should be thermogravimetry, instead of thermal gravity.
Reviewer 3 Report
Minor revision needed
- Please recheck the title of the manuscript.
- “An novel redox-active” in the abstract should be corrected.
- The starting sentence in the introduction part should be checked carefully.
- Please provide the crystal planes in the SAED image.
- Some recent works on TiO2-based materials should be incorporated in the proper places such as: 1016/j.jiec.2017.12.004; https://doi.org/10.1016/j.cej.2020.125532;
- In the experimental section, the authors should provide the purity of the reagents.
- English language should be polished throughout the manuscript.
- Scale bar of the SEM and TEM images should be clearer.
- The authors should provide the bonding states in high-resolution XPS spectra (Fig. S6, S7, and S8).
- Specify the role of Pt nanoparticles in the prepared materials.
Author Response
Please see the attchment.

Round 2
Reviewer 1 Report
After considering the revised manuscript, it requires additional comments as:
- In page 3: The two short sentences “Figure S1a showed the structure of H4TTFTB. And the preparation process of H4TTFTB-TiO2 hybrid material was illustrated in Figure S1b.” The authors should combine them into one effective sentence. Also, for the two short sentences in Page 6, line 3 from bottom “The binding energy of Ti 2p3/2 and Ti 2p1/2 is at 458.7 and 464.4 eV, respectively. And the binding energy of O 1s at 529.9 eV could be attributed to the Ti-O bond of TiO2 for a-TiO2 and H4TTFTB-TiO2-5.0 (Figure S7 and Figure S8).”
- Page 6: “For advancing the activity of photocatalytic H2 production” should be “To enhance the activity of photocatalytic H2 production”.
- Page 2/ second paragraph: The authors stated that “Tetrathiafulvalene (TTF) and its derivatives own unique electron-rich and redox-active properties” reference citation is required e.g. “https://doi.org/10.1016/S1386-1425(00)00349-8” or other reference.
